# AutoLoss: Learning Discrete Schedule for Alternate Optimization

## Abstract

Many machine learning problems involve iteratively and alternately optimizing different task objectives with respect to different sets of parameters. Appropriately scheduling the optimization of a task objective or a set of parameters is usually crucial to the quality of convergence. In this paper, we present *AutoLoss*, a meta-learning framework that automatically learns and determines the optimization schedule. AutoLoss provides a generic way to represent and learn the discrete optimization schedule from metadata, allows for a dynamic and data-driven schedule in ML problems that involve alternating updates of different parameters or from different loss objectives. We apply AutoLoss on four ML tasks: $d$-ary quadratic regression, classification using a multi-layer perceptron (MLP), image generation using GANs, and multi-task neural machine translation (NMT). We show that the AutoLoss controller is able to capture the distribution of better optimization schedules that result in higher quality of convergence on all four tasks. The trained AutoLoss controller is generalizable – it can guide and improve the learning of a new task model with different specifications, or on different datasets.

## 1 Introduction

Many machine learning (ML) problems involve iterative alternate optimization of different objectives $\{\ell_m\}_{m=1}^M$ w.r.t different sets of parameters $\{\boldsymbol{\theta}_n\}_{n=1}^N$ until a global consensus is reached. For instances, in training generative adversarial networks (GANs) (Goodfellow et al., 2014), parameters of the generator and the discriminator are alternately updated to an equilibrium; in many multi-task learning problems (Argyriou et al., 2007), one usually has to alternate the optimization of different task-specific objectives on corresponded data, until the target task performance is maximized. In these processes, one needs to determine which objective $\ell_m$ and which set of parameters $\boldsymbol{\theta}_n$ to choose at each step, and subsequently, how many iteration steps to perform for the subproblem $\min_{\boldsymbol{\theta}_n} \ell_m$. We refer to this as determining an *optimization schedule* (or *update schedule*).

While extensive research has been focused on developing better optimization algorithms or update rules (Kingma & Ba, 2014; Bello et al., 2017; Duchi et al., 2011; Sutskever et al., 2013), how to select optimization schedules has remained less studied. When the objective is complex (e.g. non-convex or combinatorial) and the parameters to be optimized are high-dimensional, the optimization schedule can directly impact the quality of convergence. However, we hypothesize that the schedule is learnable in a data-driven way, with the following empirical evidence: (1) The optimization of many ML models is sensitive to the update schedule. For examples, the updates of the generator and the discriminator in GANs are carefully reconciled to avoid otherwise model collapse or gradient vanishing (Goodfellow et al., 2014; Radford et al., 2015); In solving many multi-task learning or regularizer-augmented objectives, the optimization target $\mathcal{L}$ is a combination of multiple task-specific objectives. It is desirable to weight each objective differently as $\mathcal{L} = \sum_{m=1}^M \lambda_m \ell_m$, while different values of $\{\lambda_m\}_{m=1}^M$ result in different (local) optima. This indicates that different loss terms shall not be treated equally, and achieving the best downstream task performance requires optimizing every $\ell_m$ to different extents. (2) Previous research and practice have suggested that there do exist optimization schedules that are more probable to produce better convergence than random ones, e.g. Arjovsky et al. (2017) and Salimans et al. (2016) suggest that keeping the steps of updating the generator and discriminator of GANs at $K : 1(K > 1)$ leads to faster and more stable training of GANs.

Based on the hypothesis, in this paper, we develop *AutoLoss*, a generic meta-learning framework to automatically determine the optimization schedule in iterative and alternate optimization processes. AutoLoss introduces a parametric controller attached to an alternate optimization task. The controller

is trained to capture the relations between the past history and the current state of the optimization process, and the next step of the decision on the update schedule. It takes as input a set of status features, and decides which objectives from $\{\ell_m\}_{m=1}^M$ to optimize, and which set of parameters from $\{\boldsymbol{\theta}_n\}_{n=1}^N$ to update. The controller is trained via policy gradient to maximize the eventual outcome of the optimization (e.g. downstream task performance). Once trained, it can guide the optimization of task models to achieve higher quality of convergence faster, by predicting better schedules.

To evaluate the effectiveness of AutoLoss, we instantiate it on four typical ML tasks: $d$-ary quadratic regression, classification using a multi-layer perceptron (MLP), image generation using GANs, and neural machine translation based on multi-task learning. We propose an effective set of features and reward functions that are suitable for the controllers' learning and decisions. We show that, on all four tasks, the AutoLoss controller is able to capture the distribution of better optimization schedules that result in higher quality of convergence on the corresponding task than strong baselines. For examples, on quadratic regression with L1 regularization, it learns to detect the potential risk of overfitting, and incorporates L1 regularization when necessary, helps the task model converge to better results that can hardly be achieved by optimizing linear combinations of objective terms. On GANs, the AutoLoss controller learns to balance the training of generator and discriminator dynamically, and report both faster per-epoch convergence and better quality of generators after convergence, compared to fixed heuristic-driven schedules. On machine translation, it automatically learns to resemble human-tuned update schedules while being more flexible, and reports better perplexity results.

In summary, we make the following contributions in this paper: (1) We present a unified formulation for iterative and alternate optimization processes, based on which, we develop AutoLoss, a generic framework to learn the discrete optimization schedule of such processes using reinforcement learning (RL). To our knowledge, this is the first framework that tries to learn the optimization schedule in a data-driven way. (2) We instantiate AutoLoss on four ML tasks: $d$-ary regression, MLP classification, GANs, and NMT. We propose a novel set of features and reward functions to facilitate the training of AutoLoss controllers. (3) We empirically demonstrate AutoLoss' efficacy: it delivers higher quality of convergence for all four tasks on synthetic and real dataset than strong baselines. Training AutoLoss controller has acceptable overhead less than most hyperparameter searching methods; the trained AutoLoss controller is generalizable – it can guide and improve the training of a new task model with different specifications, or on different dataset.

## 2 RELATED WORK

**Alternate Optimization.** Many ML models are trained using algorithms with iterative and alternate workflows, such as EM (Moon, 1996), stochastic gradient descent (SGD) (Bottou, 2010), coordinate descent (Wright, 2015), multi-task learning (Zhang & Yang, 2017), etc. AutoLoss can improve these processes by learning a controller in a data-driven way, and figuring out better update schedules using this controller, as long as the schedule does affect the optimization goal. In this paper, we focus mostly on optimization problems, but note AutoLoss is applicable to alternate processes that involve non-optimization subtasks, such as sampling methods (Griffiths & Steyvers, 2004; Ma et al., 2015).

**Meta learning.** Meta learning (Andrychowicz et al., 2016; Maclaurin et al., 2015; Wang et al., 2016; Finn et al., 2017; Chen et al., 2016) has drawn considerable interest and been recently applied to improve the optimization of ML models (Ravi & Larochelle, 2016; Li & Malik, 2016; Bello et al., 2017; Fan et al., 2018). Among these works, the closest to ours are Li & Malik (2016), Bello et al. (2017), Fan et al. (2018). Li & Malik (2016) propose *learning to optimize* to directly predict the gradient values at each step of SGD. Since the gradients are continuous and usually high-dimensional, directly regressing their values might be difficult, and the learned gradient regressor is nontransferable to new models or tasks. Differently, Bello et al. (2017) propose to learn better gradient update rules based on a domain specific language. The learned rules outperform manually designed ones and are generalizable. AutoLoss differs from both – instead of learning to generate values of updates (gradients), AutoLoss focuses on producing better scheduling of updates. Therefore AutoLoss can model other classes of problems such as scheduling the generator and discriminator training in GANs (Larsen & Sønderby), or even go beyond optimization problems. In Fan et al. (2018), a *learning to teach* framework is proposed that a teacher model, trained by optimization metadata, can guide the learning of student models. AutoLoss instantiates the framework in the sense that the teacher model (controller) produces better schedules for the task model (student) optimization.

Also of note is another line of works that apply RL to enable automatic machine learning (AutoML), such as adaptive step size controller (Daniel et al., 2016), device placement optimization (Mirhoseini

et al., 2017), neural architecture search (Baker et al., 2016; Zoph & Le, 2016), etc. While addressing different problems, AutoLoss' controller is trained in a similar way (Peters & Schaal, 2008) for sequential and discrete predictions.

## 3 AUTOLOSS

**Background.** In most ML tasks, given observed data $\mathcal{D}$, we aim to minimize an objective function $\mathcal{L}(\mathcal{D}; \Theta)$ with respect to the parameters $\Theta$ of the model that we use to characterize the data. Solving this minimization problem involves finding the optima of $\Theta$ (denoted as $\Theta^*$), which we usually resort to a variety of de facto optimization methods (Boyd & Vandenberghe, 2004) if close-formed solutions are unavailable. In the rest of the paper, we will focus on two typical classes of optimization workflows which many modern ML model solvers would fall into: *iterative* and *alternate* optimization.

Iterative optimization methods look for the optimal parameter $\Theta^*$ in an *iterative-convergent* way, by repeatedly updating $\Theta$ until certain stopping criteria is reached. Specifically, at iteration $t$, the parameters $\Theta$ are updated from $\Theta^{(t)}$ to $\Theta^{(t+1)}$ following the update equation $\Theta^{(t+1)} = \Theta^{(t)} + \epsilon \cdot \Delta_{\mathcal{L}}(\mathcal{D}^{(t)}; \Theta^{(t)})$, where we denote $\Delta_{\mathcal{L}}$ as the function that calculates update values of $\Theta$ depending on $\mathcal{L}$, $\mathcal{D}^{(t)} \subseteq \mathcal{D}$ as a subset of $\mathcal{D}$ used at iteration $t$ and $\epsilon$ a scaled factor. Many widely-adopted algorithms (Bottou, 2010; Boyd et al., 2003) fall into this family, e.g. in the case for SGD, $\Delta_{\mathcal{L}}$ reduces to deriving the gradient updates $\nabla\Theta$ (we skip optional steps such as momentum or projection for clarity), $D^{(t)}$ is a stochastic batch, and $\epsilon$ is the learning rate.

To describe alternate optimization, we notice the objective $\mathcal{L}$ is usually composed of multiple different optimization targets: $\mathcal{L} = \{\ell_m\}_{m=1}^{M}$, and we want $\Theta^*$ to minimize a certain combination of them. For example, when fitting a regression model with mean square error (MSE), appending an L1 loss helps obtain sparsity; in this case, $\mathcal{L}$ is written as a linear combination of MSE and L1 terms. Similarly, the parameters $\Theta$ in many cases are also composable, e.g. when the model has multiple components with independent sets of parameters. If we decompose $\Theta = \{\theta_n\}_{n=1}^{N}$, an alternate optimization (in our definition) contains multiple steps, where each step $t$ involves choosing $\ell_{m_t} \in \mathcal{L}, \theta_{n_t} \in \Theta$, which we will call as *determining an optimization action* (notated as $a$), and update $\theta_{n_t}$ w.r.t. $\ell_{m_t}$.

Further, we note that many ML optimization tasks in practice are both iterative and alternate, such as the training process of GANs, where the updates of generator and discriminator parameters are alternated, each with a few iterations of stochastic updates, until equilibrium.

We therefore present iterative and alternate optimization with the following unified formulation:

$$\text{for } t = 1 \rightarrow T, \text{ choose } (\ell_{m_t}, \theta_{n_t}) = a_{q_t} \in \mathcal{A}, \text{ update } \theta_{n_t}^{(t+1)} = \theta_{n_t}^{(t)} + \epsilon \cdot \Delta_{\ell_{m_t}}^{n_t}, \tag{1}$$

where $\mathcal{A} = \{a_q\}_{q=1}^{Q}$ denotes the task-specific action space that defines all legitimate pairs of loss and parameter to choose from; $\Delta_{\ell_{m_t}}^{n_t}$ are update values of $\theta_{n_t}$ w.r.t. $\ell_{m_t}$. Eq. 1 reduces to the vanilla form of iterative optimization when $\mathcal{A} = \{(\mathcal{L}, \Theta)\}$.

**AutoLoss.** Given the formulation in Eq. 1, our goal is to determine $a_{q_t}$, i.e. which losses to optimize and what parameters to update at each $t$, in order to maximize the downstream task performance. We introduce a meta model, which we call *controller*, to be distinguished from the *task model* used in the downstream task. The controller is expected to learn during its exploration of task model optimization processes, and is able to decide how to update once sufficient knowledge has been accumulated.

Specifically, we let the controller make sequential decisions at each step $t$; it scans through the past history and the current states of the process (described as a feature vector $\boldsymbol{X}^{(t)} \in \mathbb{R}^K$), and predicts a one-hot vector $\boldsymbol{Y}^{(t)} \in \{0, 1\}^{|\mathcal{A}|}$, i.e. $a_q \in |\mathcal{A}|$ will be selected if the $q$th entry of $\boldsymbol{Y}^{(t)}$ is 1. We model our controller as a conditional distribution $p(\boldsymbol{y}|\boldsymbol{x}; \phi)$ parameterized by $\phi$[1], where we denote $\boldsymbol{y}$ and $\boldsymbol{x}$ as the $|\mathcal{A}|$-dim decision variable and $K$-dim feature variable, respectively. At each step $t$, we sample $\boldsymbol{Y}^{(t)} \sim p(\boldsymbol{y}|\boldsymbol{x} = \boldsymbol{X}^{(t)}; \phi)$, and perform updates following Eq. 1 and $\boldsymbol{Y}^{(t)}$.

**Parameter learning.** The parameters of the controller $\phi$ is trained to maximize the performance of the optimization task given sampled sequences of decisions within $T$ steps, notated as $\mathcal{Y} = \{\boldsymbol{Y}^{(t)}\}_{t=1}^{T}$. Accordingly, we introduce the training objective of the controller as $J(\phi) = \mathbb{E}_{\mathcal{Y} \sim p(\boldsymbol{y}|\boldsymbol{x}; \phi)}\big[R(\mathcal{Y})|\mathcal{L}, \Theta\big]$, where $R(\cdot)$ is the reward function that evaluates the final task

---

[1]The other alternative is to condition the decision at the $t$ step on the decision made at the $t - 1$ step, though we choose a simpler one to highlight the generic idea behind AutoLoss.

performance after applying the schedule $\mathcal{Y}$ for its optimization. Since the decision process involves non-differentiable sampling, we learn the parameters using REINFORCE (Williams, 1992), where the unbiased policy gradients at each updating step of the controller are estimated by sampling $S$ sequences of decisions $\{\mathcal{Y}_s\}_{s=1}^{S}$ (for all experiments we set $S = 1$) and compute

$$\nabla_\phi J(\phi) = \frac{1}{S} \sum_{s=1}^{S} \left[ (R(\mathcal{Y}_s) - B) \cdot \nabla_\phi \sum_{t=1}^{T} \log p(\boldsymbol{Y}_s^{(t)} | \boldsymbol{X}_s^{(t)}; \phi) | \mathcal{L}, \boldsymbol{\Theta} \right], \tag{2}$$

where $\boldsymbol{Y}_s^{(t)}$ is the $t$th decision in $\mathcal{Y}_s$. To reduce the variance, we introduce a baseline term $B$ in Eq. 2 to stabilize the training (similar to Pham et al. (2018)), where $B$ is defined as a moving average of received reward: $B^{(h+1)} \leftarrow \eta B^{(h)} + (1 - \eta) R^{(h)}$ with $\eta$ as a decay factor. Whenever applicable, $R(\mathcal{Y}_s) - B$ is further clipped to a given range. See detailed training algorithms in Appendix A.1.

## 4 APPLICATIONS

We next apply AutoLoss to four specific ML tasks: regression and classification with L1 regularization, image generation using GANs, and machine translation based on multi-task learning.

$d$-**ary quadratic regression and MLP classification with L1 regularization.** Given training data $\mathcal{D} = \{\boldsymbol{u}_p, v_p\}_{p=1}^{P}, \boldsymbol{u}_p \in \mathbb{R}^d, v_p \in \mathbb{R}$ generated by a linear model with Gaussian noise, we try to fit them using a $d$-ary quadratic model $f : \mathbb{R}^d \to \mathbb{R}$ as $f(\boldsymbol{u}; \boldsymbol{\Theta}) = \boldsymbol{u}^\top \boldsymbol{A} \boldsymbol{u} + \boldsymbol{b}^\top \boldsymbol{u} + c$, where parameters $\boldsymbol{\Theta} = \{\boldsymbol{A}, \boldsymbol{b}, c\}$ are optimized via minimizing the MSE $\ell_1(\boldsymbol{\Theta}) = \mathbb{E}_{(\boldsymbol{u},v)\in\mathcal{D}}[f(\boldsymbol{u}; \boldsymbol{\Theta}) - v]^2$. Since fitting the data using a higher-order model is prone to overfitting, we add an L1 term $\ell_2 = \|\boldsymbol{\Theta}\|_1$. A traditional way to find $\boldsymbol{\Theta}^*$ is to minimize $\ell_1 + \lambda \ell_2$, where $\lambda$ is a hyperparameter yet to be determined by hyperparameter search. This problem can be solved using many iterative optimization methods, e.g. SGD. To model this problem using AutoLoss, we define $\mathcal{L} = \{\ell_1, \ell_2\}$ ($M = 2$), $\boldsymbol{\Theta} = \{\boldsymbol{\theta}_1\}$ with $\boldsymbol{\theta}_1 = \{\boldsymbol{A}, \boldsymbol{b}, c\}$ ($N = 1$), and $\mathcal{A} = \{(\ell_1, \boldsymbol{\theta}), (\ell_2, \boldsymbol{\theta})\}$, i.e. the controller has Bernoulli outputs which we sample decisions from. Similarly, we apply AutoLoss in training a binary MLP classifier $f(\boldsymbol{\Theta}) : \mathbb{R}^d \to \{0, 1\}$ with ReLU nonlinearity, which is non-convex and highly prone to overfitting. We materialize $\boldsymbol{\Theta} = \{\boldsymbol{\theta}_1\}$ where $\boldsymbol{\theta}_1$ are all MLP parameters, $\mathcal{L} = \{\ell_1, \ell_2\}$ with $\ell_1$ as the binary cross entropy (BCE) and $\ell_2 = \|\boldsymbol{\Theta}\|_1$, and $\mathcal{A} = \{(\ell_1, \boldsymbol{\theta}), (\ell_2, \boldsymbol{\theta})\}$.

For both tasks, we design $\boldsymbol{X}^{(t)}$ as a concatenation of the following features in order to capture the current optimization state and the past history: (1) *training progress*: the percentile progress of training $t/T$. (2) *normalized gradient magnitude*: an $M$-dim vector where the $m$th entry is $\frac{\|\nabla_{\boldsymbol{\Theta}} \ell_m\|_2}{\sqrt{\dim(\boldsymbol{\Theta})}}$. (3) *loss values*: an $M$-dim vector $[\ell_1, \ldots, \ell_M]$ that contains values of each $\ell_m$ at $t$. Extracting features (2)(3) requires computing $\ell_m$ and $\nabla_{\boldsymbol{\Theta}} \ell_m$ repeatedly at each step $t$, which might be inefficient. We alternatively maintain and use their latest history values – we compute $\ell_m$ and $\nabla_{\boldsymbol{\Theta}} \ell_m$ only when the controller has decided to optimize $\ell_m$ at current step, and update their values in the history accordingly. (4) *validation metrics*: the loss value of $\ell_1(\boldsymbol{\Theta}^{(t)})$ (MSE for regression or BCE for classification) evaluated on a validation set, the exponential moving averages of it and of its higher-order differences. Similarly, we evaluate the validation error only when needed and use their most recent values.

For the reward function, we simply instantiated $R = \frac{C}{err}$ for regression and $R = \frac{C}{err-1}$ for classification, respectively, where $C$ is a constant, err is MSE for regression or classification error for classification, evaluated using converged parameters $\boldsymbol{\Theta}^{(T)}$ on the validation dataset. Hence, the controller obtains a larger reward if the task model achieves a lower MSE or classification error. It is worth noting that we intentionally choose these two models as a proof-of-concept that AutoLoss would work on both convex and non-convex cases. See §5 for more experiment results.

**GANs.** A vanilla GAN has two set of parameters: the parameters of the generator $G$ as $\boldsymbol{\theta}_1$ and those of the discriminator $D$ as $\boldsymbol{\theta}_2$, alternately trained via a minimax game (where $\boldsymbol{z}$ is a noise variable):

$$\min_{\boldsymbol{\theta}_1} \max_{\boldsymbol{\theta}_2} \mathcal{L}(\boldsymbol{\theta}_1, \boldsymbol{\theta}_2) = \mathbb{E}_{\boldsymbol{u}\sim p_{data}(\boldsymbol{u})}[\log D(\boldsymbol{u})] + \mathbb{E}_{\boldsymbol{z}\sim p_{\boldsymbol{z}}(\boldsymbol{z})}[\log(1 - D(G(\boldsymbol{z})))].$$

This is a typical alternate process that cannot be expressed by linear combinations of loss terms (hence can hardly benefit from hyperparameter search as in the previous two cases). How to appropriately balance the optimization of $\boldsymbol{\theta}_1$ and $\boldsymbol{\theta}_2$ is a key factor that affects the success of GAN training. Beyond fixed schedules, automatically adjusting the training of $G$ and $D$ remains rarely tackled. Fortunately, AutoLoss offers unique opportunities to learn the optimization schedules of GANs.

In particular, we instantiate $\boldsymbol{\Theta} = \{\boldsymbol{\theta}_1, \boldsymbol{\theta}_2\}$, $\mathcal{L} = \{\ell_1, \ell_2\}$ with $\ell_1 = \mathbb{E}_{\boldsymbol{z}\sim p_{\boldsymbol{z}}(\boldsymbol{z})}[\log(1 - D(G(\boldsymbol{z})))]$, $\ell_2 = -\mathbb{E}_{\boldsymbol{u}\sim p_{data}(\boldsymbol{u})}[\log D(\boldsymbol{u})] - \mathbb{E}_{\boldsymbol{z}\sim p_{\boldsymbol{z}}(\boldsymbol{z})}[\log(1 - D(G(\boldsymbol{z})))]$. To match the possible actions in

GANs training, we set $\mathcal{A}$ as $\{(\ell_1, \boldsymbol{\theta}_1), (\ell_2, \boldsymbol{\theta}_2)\}$, i.e. the controller chooses at each step to optimize one of $G$ and $D$. To track the training status of both $G$ and $D$, we reuse the same four aspects of features (1)-(4) in previous applications with the following variations: (2) We use a 3D vector $[\frac{\|\nabla_{\boldsymbol{\theta}_1}\ell_1\|_2}{\sqrt{\dim(\boldsymbol{\theta}_1)}}, \frac{\|\nabla_{\boldsymbol{\theta}_2}\ell_2\|_2}{\sqrt{\dim(\boldsymbol{\theta}_2)}}, \log\frac{\|\nabla_{\boldsymbol{\theta}_1}\ell_1\|_2 \cdot \sqrt{\dim(\boldsymbol{\theta}_2)}}{\|\nabla_{\boldsymbol{\theta}_2}\ell_2\|_2 \cdot \sqrt{\dim(\boldsymbol{\theta}_1)}}]$, where the first two entries are gradient norms of $G$ and $D$, respectively, while the third is their log ratio to reflect how balanced the updates are; (3) A vector of training losses and their ratio $[\ell_1, \ell_2, \frac{\ell_1}{\ell_2}]$; (4) As there is no clear validation metric to evaluate a GAN, for $G$, we generate a few samples given its current state of parameters $\boldsymbol{\theta}_1^{(t)}$, and compute the *inception score* (notated as $\mathcal{IS}$) of them as a feature to indicate how good $G$ is. For $D$, we sample equal number of samples from both $G$ and the training set and use $D$'s classification error (classified as real or fake) on them as a feature. For (2)-(4), we similarly use their most recent history values for improved efficiency. In a same way, we instantiate $R = C \cdot \mathcal{IS}^2$ to encourage the controller to predict schedules that lead to better generators.

**Multi-task machine translation.** Most multi-task learning problems require optimizing several domain-specific objectives jointly for improved performance (Argyriou et al., 2007). However, without carefully weighting or scheduling of the optimization of each objective, the results may unexpectedly degrade than optimizing a single objective (Zhang & Yang, 2017; Teh et al., 2017). As the third task, we apply AutoLoss to find better optimization schedules for multi-task learning based neural machine translation (NMT). Following Niehues & Cho (2017), we build an attention-based encoder-decoder model with three task objectives: the target task translates German into English ($\ell_1$), while the secondary tasks are German named entity recognition (NER) ($\ell_2$) and German POS tagging ($\ell_3$). We use a shared encoder $E$ with parameters $\boldsymbol{\theta}_e$ and separate decoders $D_{MT}, D_{NER}, D_{POS}$ with parameters as $\boldsymbol{\theta}_d^{MT}, \boldsymbol{\theta}_d^{NER}, \boldsymbol{\theta}_d^{POS}$, respectively. To fit within AutoLoss, we set $\mathcal{L} = \{\ell_m\}_{m=1}^3$, $\Theta = \{\boldsymbol{\theta}_1, \boldsymbol{\theta}_2, \boldsymbol{\theta}_3\}$ with $\boldsymbol{\theta}_1 = \{\boldsymbol{\theta}_e, \boldsymbol{\theta}_d^{MT}\}, \boldsymbol{\theta}_2 = \{\boldsymbol{\theta}_e, \boldsymbol{\theta}_d^{NER}\}, \boldsymbol{\theta}_3 = \{\boldsymbol{\theta}_e, \boldsymbol{\theta}_d^{POS}\}$, and the action space $\mathcal{A} = \{(\ell_1, \boldsymbol{\theta}_1), (\ell_2, \boldsymbol{\theta}_2), (\ell_3, \boldsymbol{\theta}_3)\}$, i.e. we optimize a task at a time. Still, we reuse the same set of features in previous tasks with small revisions, and set the reward function $R = C \cdot \text{PPL}$ where PPL is the validation perplexity. More details about the NMT task are provided in Appendix A.3.

**Discussion.** When the task model is complex and requires numerous iterations to converge (i.e. when $T$ in Eq. 2 is large), the controller receives sparse and delayed rewards. To facilitate the training, we adapt $T$ depending on the task: for simpler tasks that converge with fewer iterations (e.g. regression and MLP classification), $T$ equals the number of steps to convergence. For the NMT task that needs longer exploration, we set $T$ as a fixed constant (instead of the max number to convergence) and online train the controller using proximal policy optimization (PPO) algorithm (see Appendix A.1)[2]. We accordingly adjust the reward function as $R^{(t)\to(t+T)} = C \cdot (P^{(t+T)} - P^{(t)})/(\frac{P^{(t)} - P^{(t-kT)}}{k})$ where $k$ is a hyperparameter and $P$ is PPL for NMT, i.e. we generate a reward every $T$ steps based on the improvement of performance and use it as reward for each step in this segment of steps. Since the improvement will be tiny around optima, we normalize the reward by dividing $\frac{P^{(t)} - P^{(t-kT)}}{k}$ in case the reward is too small to provide enough training signal.

## 5 EVALUATION

In this section, we evaluate AutoLoss empirically on the four tasks using synthetic and real data.

### 5.1 QUALITY OF CONVERGENCE

We first verify the feasibility of the AutoLoss idea. We empirically show that under the formulation of Eq. 1, there *do exist* learnable update schedules, and AutoLoss is able to capture their distribution and guides the task model to achieve better quality of convergence across multiple tasks and models.

**Regression and classification with L1 regularization.** We first apply AutoLoss on two relatively simple tasks with synthetic data, and see whether it can outperform its alternatives (e.g. minimizing linear combinations of loss terms) in combating overfitting. Specifically, for regression, we synthesize dataset $\mathcal{D} = \{\boldsymbol{u}_p, v_p\}_{p=1}^P$ using a linear model with Gaussian noise (in the form of $v = \boldsymbol{w} \cdot \boldsymbol{u} + \xi$). In this case, a quadratic regressor is over-expressive and highly likely to overfit the data if without proper regularization. Similarly, for MLP, we synthesize a classification dataset with risks of overfitting by letting only $5\%$ dimensions in $\boldsymbol{u}$ be informative whereas the rest be either linear combinations of them or random noise. Details of how the data are synthesized are provided in the Appendix A.2.

---

[2]Note that this online training strategy introduces the "short-horizon bias" (Wu et al., 2018). Empirically, we observe this bias affects the GAN task compared to offline training, but is insignificant in the NMT task.

| Metric | W/O L1 | S1 | S2 | S3 | DGS | AUTOLOSS |
|--------|--------|-----|-----|-----|-----|----------|
| *MSE* | .790 (1e-5) | .086 (2e-3) | .096 (2e-3) | .095 (1e-3) | .086 (3e-4) | **.070 (1e-3)** |
| *err* | .124 (3e-5) | .091 (1e-3) | .094 (2e-3) | .094 (2e-3) | .093 (2e-6) | **.088 (2e-3)** |

Table 1: AUTOLOSS vs. W/O L1, schedules S1-S3, and DGS on two tasks. Results are averaged over 10 trials. We substitute a baseline MSE (3.94) from the results caused by noise during data generation.

We split our dataset into 5 parts following Fan et al. (2018): $\mathcal{D}_{train}^C$ and $\mathcal{D}_{val}^C$ for controller training; Once trained, the controller is used to guide the training of a new task model on another two partitions $\mathcal{D}_{train}^T, \mathcal{D}_{val}^T$. Hence, the controller would not work by just memorizing good schedules on $\mathcal{D}_{train}^C$. We reserve the fifth partition $\mathcal{D}_{test}$ to assess the task model after guided training. For both regression and classification, our controller is simply a two-layer MLP with ReLU activation.

We compare MSE or classification error (err) evaluated on $\mathcal{D}_{test}$ in Table 1 to the following methods: (1) W/O L1: which minimizes only an MSE or BCE term on $\mathcal{D}_{train}^T \cup \mathcal{D}_{val}^T$. (2) We designed three flexible schedules that optimize the L1 term at each iteration if the condition $\frac{A-B}{B} > th$ is met, where $A, B$ are (S1) task loss values ($\ell_1$) evaluated on $\mathcal{D}_{val}^T$ and $\mathcal{D}_{train}^T$ respectively, (S2) L1 loss and task loss evaluated on $\mathcal{D}_{val}^T$, (S3) gradient norms of L1 and MSE loss. We grid search the threshold $th$ on training data and only report best achieved results. (3) DGS: we minimize $\ell_1 + \lambda\ell_2$ with $\lambda$ determined by dense grid search (DGS); Particularly, we densely grid search the best $\lambda$ from a pre-selected interval using 50 experiments, and report the best MSE.[3]

Without regularization, the performance deteriorates – we observed the large gap between W/O L1 and others with L1 on both tasks (convex and non-convex). AutoLoss manages to detect and combat the potential risk of overfitting with the designed features, and automatically optimizes the provided L1 term when appropriate. In terms of task performance, AUTOLOSS outperforms three manually designed schedules as well as DGS, a practically very strong method.

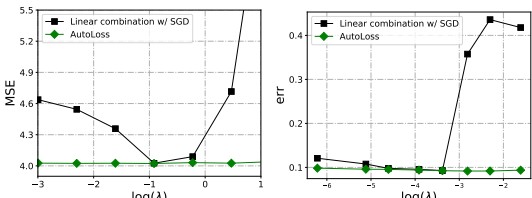

Figure 1: AutoLoss reaches good convergence regardless of $\lambda$ for both regression and MLP classification.

This is not unexpected as AutoLoss' parametric controller offers more flexibility than heuristic-driven schedules, or any fixed-formed objectives with a dense grid of $\lambda$ values (i.e. DGS). To understand this, consider the $d$-ary quadratic regression which is convex and has global optima only determined by $\lambda$. AutoLoss frees the loss surface from being strictly characterized in the form of a linear combination equation, thus allows for finding better optimal solutions that not only enjoy the regularizer effects (i.e. sparsity), but also more closely characterize observed data. As a side benefit, AutoLoss liberates us from hyper-searching $\lambda$, which might be difficult or expensive, and not transferable from one model/dataset to another. We perform an additional experiment in Figure 1 where we set different $\lambda$ in $\ell_2 = \lambda|\mathbf{\Theta}|_1$, and note AutoLoss always reaches the same quality of convergence regardless of $\lambda$. Similar results are observed on MLP classification, a highly non-convex model. The results suggest AutoLoss might be a better alternative to incorporate regularization than fixed-formed combinations of loss terms. We further provide an ablation study on the importance of each designed feature in the Appendix A.4.

**GANs.** We next use AutoLoss to help train GANs to generate images. We first build a DCGAN with the architecture of $G$ and $D$ following Radford et al. (2015), and train it on MNIST. As the task model itself is hard to train, in this experiment, we set the controller as a linear model with Bernoulli outputs. GAN's minimax loss goes beyond the form of linear combinations, and there is no rigorous evidence showing how the training of $G$ and $D$ shall be scheduled. Following common practice, we compare AUTOLOSS to the following baselines: (1) GAN: the vanilla GAN where $D$ and $G$ are alternately updated once a time; (2) GAN 1:K: suggested by some literature, we build a series of baselines that update $D$ and $G$ at the ratio 1:K (K = 3, 5, 7, 9, 11) in case $D$ is over-trained to reject all samples by $G$; (3) GAN K:1: that we contrarily bias toward more updates for $D$. To evaluate $G$, we use the inception score ($\mathcal{IS}$) (Salimans et al., 2016) as a quantitative metric, and also visually inspect generated results. To calculate $\mathcal{IS}$ of digit images, we follow Deng et al. (2017) and use a trained CNN classifier on MNIST train split as the "inception network" (real MNIST images have $\mathcal{IS} = 9.5$ on it). In Figure 2, we plot the $\mathcal{IS}$ w.r.t. number of training epochs, comparing AUTOLOSS to four best performed baselines out of all GAN (1:K) and GAN (K:1), each with three trials of

---

[3]Note that the DGS presented is a very strong baseline and might even be unrealistic in practice due to unacceptable cost or lack of prior knowledge on hyperparameters.

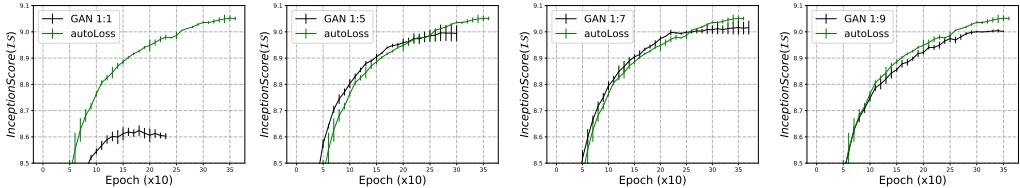

Figure 2: AUTOLOSS vs. 4 best performed baselines in terms of training progress ($\mathcal{IS}$ vs. epochs). Each curve corresponds to 3 runs of experiments and the variances are illustrated as vertical bar.

experiments. We also report the converged $\mathcal{IS}$ for all methods here: 8.6307, 9.0026, 9.0232, 9.0145, **9.0549** for GAN, GAN (1:5), GAN (1:7), GAN (1:9), AUTOLOSS, respectively.

In general, GANs trained with AutoLoss present higher quality of final convergence in terms of $\mathcal{IS}$ than all baselines. For example, comparing to GAN 1:1, AUTOLOSS improves the converged $\mathcal{IS}$ for 0.5, and is almost 3x faster to achieve where GAN 1:1 converges ($\mathcal{IS} = 8.6$) in average. We observe GAN 1:7 performs closest to AUTOLOSS: it achieves $\mathcal{IS} = 9.02$, compared to AUTOLOSS 9.05, but exhibits higher variance in multiple experiments. It is worth noting that all GAN K:1 baselines perform worse than the rest and are skipped in Figure 2. We visualize some generated digit images by AutoLoss-guided GANs in the Appendix A.6 and find the visual quality directly relevant with $\mathcal{IS}$ and no mode collapse is observed.

**Multi-task machine translation.** Lastly, we evaluate AutoLoss on multi-task NMT. Our NN architecture exactly follows the one in Niehues & Cho (2017). More information about the dataset and experiment settings are provided in Appendix A.3 and Niehues & Cho (2017). We use an MLP controller with a 3-way softmax output, and train it along with the NMT model training, and compare it to the following approaches: (1) MT: single-task NMT baseline trained with parallel data; (2) FIXEDRATIO: a manually designed schedule that selects which task objective to optimize next based on a ratio proportional to the size of training data for each task; (3) FINETUNED MT: train with FIXEDRATIO first and then fine-tune delicately on MT task. Note that baselines (2) and (3) are searched and heavily tuned by authors of Niehues & Cho (2017). We evaluate the perplexity (PPL) on validation set w.r.t. training epochs in Fig 3(L), and report the final converged PPL as well: 3.77, 3.68, 3.64, **3.54** for MT, FIXEDRATIO, FINETUNED MT and AUTOLOSS, respectively.

We observe that all methods progress similarly but AUTOLOSS and FINETUNE MT surpass the other two after several epochs. AUTOLOSS performs similarly to FINETUNE MT in training progress before epoch 10, though AUTOLOSS learns the schedule fully automatically while FINETUNE MT requires heavy manual crafting. AutoLoss is about 5x faster than FIXEDRATIO to reach where the latter converges, and reports the lowest PPL than others after convergence, crediting to its higher flexibility. We visualize the controller's softmax output after convergence in Fig 3(M). It is interesting to notice that the controller meta-learns to up-weight the target NMT objective at later phase of the training. This, in some sense, seems to resembles the "fine-tuning the target task" strategy appeared in many multi-task learning literature, but is much more flexible thanks to the parametric controller.

**Overhead.** AutoLoss introduces three possible sources of overheads: controller feature extraction, controller inference and training, and potential cost by additional task model training. Since we build features merely based on existing metadata or histories (see §4), which have to be computed anyway

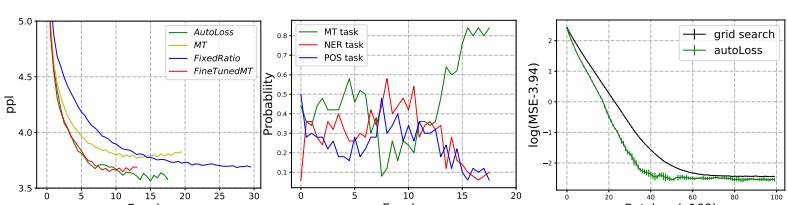

Figure 3: (L) Validaton PPL w.r.t. training epochs on the NMT task; (M) Visualization of the trained controller's policy on the NMT task; (R) AUTOLOSS vs. DGS in terms of MSE w.r.t. scanned batches on the regression task.

even without AutoLoss, the feature extraction has negligible overhead. Moreover, as a simple 2-layer MLP controller would suffice for many applications per our experiments, training or inference with the controller add minimal computational overhead, especially on modern hardware such as GPUs. Besides, for tasks that converge shortly within a few iterations (e.g. $d$-ary regression and MLP classification), AutoLoss, similar to grid search, requires repeating multiple experiments in order to accumulate sufficient supervisions ($T$ is # of steps to converge). To assess the resulted overhead, we perform a fixed budget experiment: given a fixed number of data batches allowed to scan, we

compare in Fig 3(R) the reached convergence by AUTOLOSS and DGS on the regression task. We observe AutoLoss is much more sample-efficient – it achieves better convergence with less training runs. On the other hand, for computational-heavy tasks that need many steps to converge (GANs, NMT), the controller training, in most cases, can finish simultaneously with task model training, and does not repeat experiments as many times as other hyperparameter search methods would do.

## 5.2 TRANSFERABILITY

We next investigate the transferability of a trained controller to different models or datasets.

**Transfer to different models.** To see whether a differently configured task model can benefit from a trained controller, we design the following experiment: we let a trained DCGAN controller on MNIST guide the training of new GANs (from scratch) whose $G$ and $D$ have randomly sampled neural architectures. We describe the sampling strategies in Appendix A.5. We compare the (averaged) converged $\mathcal{IS}$ between with and without the AutoLoss controller in Fig 4b, while we skip cases that both AutoLoss and the baseline fail ($\mathcal{IS} < 6$) because improper neural architectures are sampled. AU-TOLOSS manages to generalize to unseen architectures, and outperforms DCGAN in 16 out of 20 architectures. This proves that the trained controller is not simply memorizing the optimization behavior of the specific task model it is trained with; instead, the knowledge learned on a model is generalizable to novel models.

| Dataset # | W/O L1 | DGS | AUTOLOSS |
|-----------|--------|--------|----------|
| 1 | .1337 | **.1019** | .1037 |
| 2 | .1294 | .1035 | **.1016** |
| 3 | .1318 | .1022 | **.0997** |

(a)

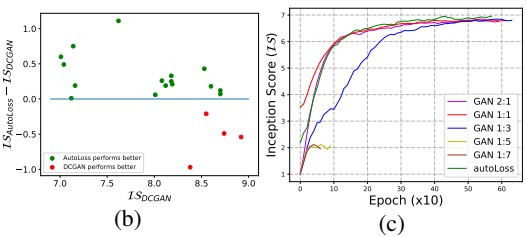

(b)          (c)

Figure 4: (a) Transfer a trained controller for MLP classification to different data distributions. (b) Comparing the final convergence ($\mathcal{IS}$) on training randomly sampled DCGAN architectures w/ and w/o AutoLoss. (c) Comparing the training progress ($\mathcal{IS}$ vs. epochs) of GAN 1:K, GAN K:1 and an AutoLoss-guided GAN on CIFAR-10 with the controller trained on MNIST.

**Transfer to different data distributions.** Our second set of experiments try to figure out whether an AutoLoss controller can generalize to different data distributions. Accordingly, we let a trained controller on one dataset to guide the training of the same task model from scratch, but on a different dataset with totally different distributions. We compare the AutoLoss-trained model to other methods, and report the results in Figure 4a and Figure 4c on two tasks: MLP classification, for which we synthesize 4 datasets following a generative process with 4 different specifications (therefore different distributions), with one of them used for controller training; GANs, where we first train a controller for digit generation on MNIST, and use the controller to guide the training of the same GAN architecture on CIFAR-10. In both cases, we observe AutoLoss manages to guide the model training on unseen data. On MLP classification, it delivers trained models comparable to or better than models searched via DGS, while being 50x more economical – DGS has to repeat 50 or more experiments to achieve the reported results on unseen data or model, while AutoLoss, once trained, is free at inference phase. On image generation, when transferred from digit images to natural images, a controller guided GAN achieves both higher quality of convergence and faster per-epoch convergence than a normal GAN trained with various fixed schedules, among which we observe GAN 1:1 performs best on CIFAR-10, while most of GAN K:1 schedules fail. We visually inspect the images generated by DCGANs guided by AutoLoss and find the image quality satisfying and no mode collapse occurred, with converged $\mathcal{IS} \approx 7$, compared to best reported $\mathcal{IS} = 6.16$ by DCGAN in previous literature. Visualization of the generated CIFAR-10 images can be found in Appendix A.7.

We are also interested in knowing whether a trained controller is transferable when both data and models change. We transfer a DCGAN controller trained on MNIST to a different DCGAN on CIFAR-10, and observe comparable quality and speed of convergence to the best fixed schedule on CIFAR-10, though AutoLoss bypasses the schedule search and is more readily available.

## 6 CONCLUSION

We propose a unified formulation for iterative alternate optimization and developed AutoLoss, a framework to automatically learn and generate optimization schedules. Comprehensive experiments on synthetic and real data have demonstrated that the optimization schedule produced by AutoLoss controller can guide the task model to achieve better quality of convergence, and the trained AutoLoss controller is transferable from one dataset to another, or one model to another.

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

## A Appendix

### A.1 Training Algorithms

For all our experiments, we set $S = 1$. For simple tasks such as d-ary regression and MLP classification that converge quickly in a few steps (therefore less costly), we set $T$ as the number of iterations took for a training instance to converge, i.e. a reward is generated upon the completion of a training instance, and we repeat multiple training instances until the controller has converged. For computational-heavy tasks that require many iterations to converge, we formulate the episodic scenery into continuous scenery by concatenating training instances together and set discount rate $\gamma$ to 0.95. We set $T = 20$, meaning that we evaluate $R$ to generate an intermediate reward every $T$ steps (before convergence), and perform a PPO update step, in case the exploration takes too long and the reward is too sparse.

---

**Algorithm 1** Training AutoLoss controller along with a task model (offline version).

---

1: Determine task-specific parameters $S$ and $T$.
2: **repeat**
3:     **for** $s = 1 \rightarrow S$ **do**
4:         **for** $t = 1 \rightarrow T$ **do**
5:             Extracting the optimization state feature $\boldsymbol{X}^{(t)}$.
6:             Determine the optimization action $a_{q_t} = (\ell_{m_t}, \boldsymbol{\theta}_{n_t})$ by sampling $\boldsymbol{Y}^{(t)} \sim p(\boldsymbol{y}|\boldsymbol{x} = \boldsymbol{X}^{(t)}; \boldsymbol{\phi})$.
7:             Perform one step of the task model optimization: $\boldsymbol{\theta}_{n_t} \leftarrow \boldsymbol{\theta}_{n_t} + \epsilon \cdot \Delta^{n_t}_{\ell_{m_t}}$
8:         **end for**
9:         Evaluate the reward $R(\mathcal{Y}_s) = R(\{\boldsymbol{Y}^{(t)}\}_{t=1}^T)$ received so far, and generate a pair $(\mathcal{Y}_s, R(\mathcal{Y}_s))$.
10:     **end for**
11:     Update controller parameters $\phi$ using Eq. 2 and $\{(\mathcal{Y}_s, R(\mathcal{Y}_s))\}_{s=1}^S$
12: **until** convergence.

---

---

**Algorithm 2** Training AutoLoss controller along with a task model (online version).

---

1: Determine task-specific parameters $S$, $T$ and $K$.
2: Evaluate the performance of initialized task model on validation set $P^{(0)}$ (PPL for NMT)
3: Initialize the moving average of performance improvement $\overline{\Delta P}$
4: **repeat**
5:     **for** $s = 1 \rightarrow S$ **do**
6:         **for** $t = 1 \rightarrow T$ **do**
7:             Extracting the optimization state feature $\boldsymbol{X}^{(t)}$.
8:             Determine the optimization action $a_{q_t} = (\ell_{m_t}, \boldsymbol{\theta}_{n_t})$ by sampling $\boldsymbol{Y}^{(t)} \sim p(\boldsymbol{y}|\boldsymbol{x} = \boldsymbol{X}^{(t)}; \boldsymbol{\phi}_{old})$.
9:             Perform one step of the task model optimization: $\boldsymbol{\theta}_{n_t} \leftarrow \boldsymbol{\theta}_{n_t} + \epsilon \cdot \Delta^{n_t}_{\ell_{m_t}}$
10:         **end for**
11:         Evaluate the performance of task model $P^{(T)}$ on validation set after T steps of training.
12:         Calculate the reward $R^{(0) \rightarrow (T)} = C \cdot (P^{(0)} - P^{(T)})/\overline{\Delta P}$
13:         Generate a sequence of transitions $\{\boldsymbol{X}^{(t)}, \boldsymbol{Y}^{(t)}, R^{(0) \rightarrow (T)}, \boldsymbol{X}^{(t+1)}\}_{t=1}^{T-1}$ and add them into replay buffer.
14:         $\overline{\Delta P} \leftarrow \lambda \overline{\Delta P} + (1 - \lambda)(P^{(0)} - P^{(T)})$
15:         $P^{(0)} \leftarrow P^{(T)}$
16:     **end for**
17:     Update controller parameters $\phi$ using PPO for $K$ times with minibatches randomly sampled from replay buffer.
18:     $\phi_{old} \leftarrow \phi$
19: **until** convergence.

---

### A.2 Data Synthesis for $d$-ary Quadratic Regression and MLP Classification

For the experiments in §5.1, we generate the dataset $\mathcal{D} = \{\boldsymbol{u}_p, v_p\}_{p=1}^P$ for the $d$-ary quadratic regression task as follows:

- Sample the weight vector $\boldsymbol{w} \sim \text{Uniform}[-0.5, 0.5]$.

| Feature to drop | MSE |
|---|---|
| (2) normalized gradient magnitude | .086 |
| (3) loss values | .101 |
| (4) validation metrics | .085 |
| None | **.070** |

Table 2: The MSE performance when some features presented in §4 are ablated on the regression task.

- For $p = 1 \rightarrow P$:
    - Sample the feature vector $\boldsymbol{u}_p \sim \text{Uniform}[-5, 5]$.
    - Sample a Gaussian noise $\xi_p \in \mathcal{N}(0, 2)$.
    - Generate $v_p = \boldsymbol{w}^T \cdot \boldsymbol{u}_p + \xi_p$.

For the MLP classification task, we synthesize the data $\mathcal{D} = \{\boldsymbol{u}_p, v_p\}_{p=1}^P$ as follows.

- Create four cluster centers $\{\mathcal{C}^c\}_{c=1}^4$ by sampling from the vertices of a hypercube.
- Assign two centers $\mathcal{C}^1, \mathcal{C}^2$ as positive ($v = 1$) while $\mathcal{C}^3, \mathcal{C}^4$ as negative ($v = 0$).
- For $p = 1 \rightarrow P$:
    - Sample the label $v_p \sim \{0, 1\}$.
    - Sample $\mathcal{C}$ from $\{\mathcal{C}^1, \mathcal{C}^2\}$ if $v_p = 1$ otherwise $\{\mathcal{C}^3, \mathcal{C}^4\}$.
    - Sample $\xi_p \sim \mathcal{N}(0, 1)$ and generate a vector $\boldsymbol{u}_p^1 = \mathcal{C} + \xi_p$ as the first 5% dimensions of $\boldsymbol{u}_p$.
    - Generate $\boldsymbol{u}_p^2$ as another 5% dimensions of $\boldsymbol{u}_p$, by randomly linearly combining the dimensions in $\boldsymbol{u}_p^1$,
    - Generate $\boldsymbol{u}_p^3$ as the rest dimensions of $\boldsymbol{u}_p$, by sampling from $\mathcal{N}(0, 1)$.
    - Generate $\boldsymbol{u}_p = [\boldsymbol{u}_p^1, \boldsymbol{u}_p^2, \boldsymbol{u}_p^3]$

## A.3 MULTI-TASK MACHINE TRANSLATION DATA AND ARCHITECTURE

### A.3.1 DATA

For the translation task, we use WIT corpus (Cettolo et al., 2012) for German to English translation. To accelerate training, we only use one fourth of all data, which has 1M tokens. For the POS tagging task, we use the Tiger Corpus (Brants et al., 2004). The POS tag set consists of 54 tags. The German named-entity tagger is trained on GermEval 2014 NER Shared Task data (Benikova et al., 2014). The corpus is extracted from Wikipedia and tag set consists of 24 tags.

We preprocess the data by tokenizing, true-casing and replacing all Arabic number by zero. In addition, we apply byte-pair encoding with 10K subwords on source and target side of the WIT corpus separately. We then apply the subwords to all German and English corpora.

### A.3.2 ARCHITECTURE

For the task model, we use an attentional encoder-decoder architecture. The three tasks share one encoder $E$ and have their own decoders $D_{MT}, D_{NER}, D_{POS}$. The encoder is a two-layer bidirectional LSTM with 256 hidden units. All decoders are also two-layer bidirectional LSTMs with luong attention (Luong et al., 2015) on the top layer. All hidden sizes in decoders are 256. The word embeddings have a size of 128.

For the controller model, instead of REINFORCE, we apply Proximal Policy Optimization algorithm (PPO) (Schulman et al., 2017) to train the controller. Both actor net and critic net are two-layer perceptrons with hidden size 32.

## A.4 FEATURE ABLATION STUDY

We investigate the importance of the designed controller features presented in §4. In particular, we report in Table 2 the performance on the regression task after dropping one of the features, where we

find that all features being useful while feature (3), which captures the most recent values of all loss terms, bringing the biggest improvement. We also tried current and historical states of parameters, gradients, and momentums, and found the set of features presented in §4 achieve best trade-off on performance and efficiency.

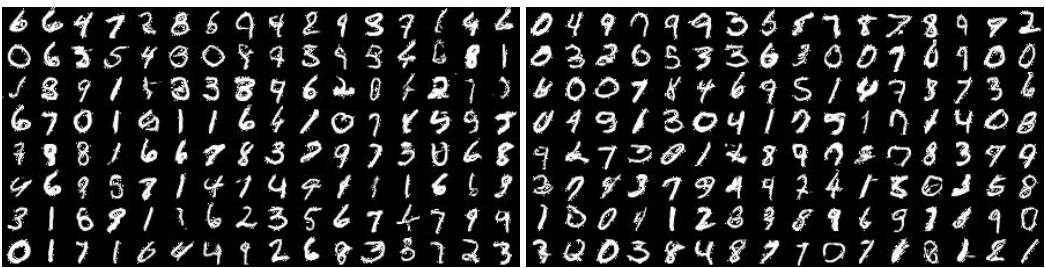

Figure 5: Images generated by a DCGAN trained under AutoLoss' controller on MNIST. The controller is trained along with the DCGAN training.

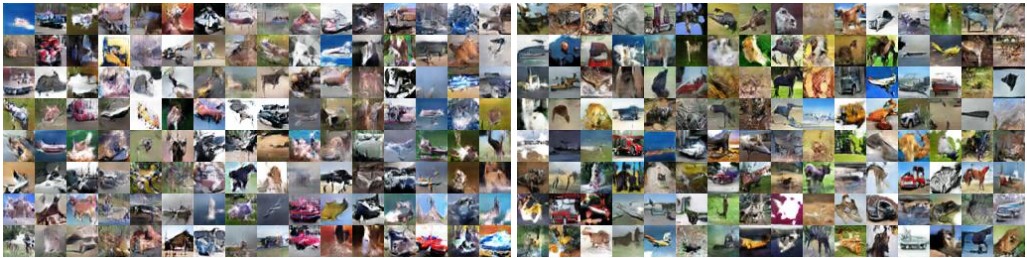

Figure 6: Images generated by an AutoLoss-guided DCGAN. The controller is trained on MNIST dataset and applied to guide the training of GANs on CIFAR-10.

### A.5   Sample Strategies to Generate Random DCGAN Architectures

For the experiments in §5.2, we generate DCGAN architectures (Radford et al., 2015; Salimans et al., 2016) by randomly sampling the following configurations:

- Sample the number of filters in the base layer of $G$ and $D$ from $\{32, 64, 128\}$.
- Sample dim$(z)$ from $\{64, 128\}$.
- Decide whether to use batchnorm or not.
- Sample the activation functions from $\{\text{ReLU}, \text{LeakyReLU}\}$.

This results in $3 \times 2 \times 2 \times 2 = 24$ possible DCGAN architectures, among which some of them fail to converge during its training according to our experiments.

### A.6   Image Generated by AutoLoss-guided GANs on MNIST

In addition to §5.1, we illustrate in Fig 5 the digit images generated by DCGANs trained under AutoLoss' policy, which report the highest $\mathcal{IS} = 9.0549$ on MNIST than any other fixed schedule.

### A.7   CIFAR-10 images generated by GANs guided by an AutoLoss controller trained on MNIST

In Fig 6, we illustrate some images generated by DCGANs under guided training of an AutoLoss controller trained on MNIST. We observe the visual quality of generated images are reasonably good.

## A.8 TRAINING PARAMETERS FOR EACH TASK

$d$**-ary quadratic regression.** For the task model, we use Adam optimizer with learning rate 0.0005 and batch size 50. Early stop is applied with an endurance of 100 batches. The controller is trained via Algorithm 1, with Adam optimizer and learning rate 0.001.

**MLP classification.** For the task model, we use Adam optimizer with learning rate 0.0005 and batch size 200. Early stop is applied with an endurance of 100 batches. The controller is trained via Algorithm 1, with Adam optimizer and learning rate 0.001.

**GANs.** For the task model, we use Adam optimizer with learning rate 0.0002 and batch size 128. Early stop is applied with an endurance of 20 epochs. The controller can be trained either via Algorithm 1 or Algorithm 2. Empirically, we observe Algorithm 1 produces best results with Adam optimizer and learning rate at 0.001.

**NMT.** For the task model, we use Adam optimizer with learning rate 0.0005 and dropout rate 0.3 at each layer. All gradients are clipped within 1. Batch size is 128. The controller can be trained either via Algorithm 1 or Algorithm 1. The best performed controller is trained by Algorithm 2, where we use Adam optimizer with learning rate 0.001, buffer size 2000 and batch size 64.

## A.9 LIMITATIONS AND FUTURE WORK

While Autoloss offers a generic way to parameterize and learn the optimization schedule, we observe it exhibits the following limitations during our development of this framework, and leave some of them as future works.

**Bounded transferability.** We observe AutoLoss has bounded transferability – while we successfully transfer a controller across different CNNs, we can hardly transfer a controller trained for CNNs to RNNs. This is slightly different from some related AutoML works, such as in (Bello et al., 2017), where auto-learned neural optimizers are able to produce decent results on even different families of neural networks. We hypothesize that the optimization behaviors or trajectories of CNNs and RNNs are very different, hence the function mappings from status features to actions are different. We leave it as a future work to study where the clear boundary is.

**Design whitebox features to capture optimization status.** Another limitation of AutoLoss is the necessity of designing the feature vector $X$, which might require some prior knowledge on the task of interest, such as being aware of a rough range of the possible values of validation metrics, etc. In fact, We initially experimented with directly feeding blackbox features (e.g. raw vectors of parameters, gradients, momentum, etc.) into controller, but found they empirically contributed little to the prediction, and sometimes hindered transferability (as different models have their parameter or gradient values at different scales).

**Non-differentiable optimization.** Meta-learning discrete schedules involves non-differentiable optimization, which is by nature difficult. Therefore, a lot of techniques in addition to vanilla REINFORCE are required to stabilize the training. As a potential future work, we will seek for continuous representations of the update schedules and end-to-end training methodologies, as arisen in recent works (Liu et al., 2018).

