# OpenReview forum: "AutoLoss: Learning Discrete Schedule for Alternate Optimization"
_ICLR.cc/2019/Conference_

### Official Review · AnonReviewer3 · 2018-11-05
**Nice work. Will the work be open source ?**

**Rating:** 7
**Confidence:** 3

**Review:**

This paper addresses a novel variant of AutoML, to automatically learn and generate optimization schedules for iterative alternate optimization problems. The problem is formulated as a RL problem, and comprehensive experiments on four various applications have demonstrated that the optimization schedule produced can guide the task model to achieve better quality of convergence, more sample-efficient, and the trained controller is transferable between datasets and models. Overall, the writing is quite clear, the problem is interesting and important, and the results are promising.

Some suggestions:

1. What are the key limitations of AutoLoss ? Did we observe some undesirable behavior of the learned optimization schedule, especially when transfer between different datasets or different models ? More discussions on these questions can be very helpful to further understand the proposed method.

2. As the problem is formulated as an RL problem, which is well-known for its difficulty in training, did we encounter similar issues? More details in the implementation can be very helpful for reproducibility.

3. Any plan for open source ?

---

> ### Author Response · Authors · 2018-11-13
> **Response to AnonReviewer3**
>
> Thank you for the valuable and encouraging feedback! Below, please see our replies.
>
> >> What are the key limitations of AutoLoss? Did we observe some undesirable behavior of the learned optimization schedule, especially when transfer between different datasets or different models ? More discussions on these questions can be very helpful to further understand the proposed method.
>
> These are indeed good questions. We list several limitations we discovered during the development of AutoLoss:
> - Bounded transferability
> We observe AutoLoss has bounded transferability -- while we successfully transfer a controller across different CNNs, we can hardly transfer a controller trained for CNNs to RNNs. This is slightly different from some related AutoML works, such as in [1], where auto-learned neural optimizers are able to produce decent results on even different families of neural networks. We hypothesize that the optimization behaviors or trajectories of CNNs and RNNs are very different, hence the function mappings from status features to actions are different. We leave it as a future work to study where the clear boundary is.
> - Design white-box features to capture optimization status
> Another limitation of AutoLoss is the necessity of designing the feature vector X, which might require some prior knowledge on the task of interest, such as being aware of a rough range of the possible values of validation metrics, etc. In fact, We initially experimented with directly feeding blackbox features (e.g. raw vectors of parameters, gradients, momentum, etc.) into controller, but found they empirically contributed little to the prediction, and sometimes hindered transferability (as different models have their parameter or gradient values at different scales).
> - Non-differentiable optimization
> Meta-learning discrete schedules involves non-differentiable optimization, which is by nature difficult. Therefore, a lot of techniques in addition to vanilla REINFORCE are required to stabilize the training. Please also see our answer to the next question for more details.
> As a potential future work, we will seek for continuous representations of the update schedules and end-to-end training methodologies, as arisen in recent works [2].
>
> We haved add the above discussion to the latest version as Appendix A.9.
>
> >> As the problem is formulated as an RL problem, which is well-known for its difficulty in training, did we encounter similar issues? More details in the implementation can be very helpful for reproducibility.
> >> Any plan for open source?
>
> We acknowledge the difficulties of training controllers using vanilla REINFORCE. During our development of the training algorithm (See Eq.2, the “discussion” section in Sec.4, and Appendix A.1), we found the vanilla form of REINFORCE algorithm leads to unstable training. We therefore have made many improvements and adaptations by either referring to existing literature, or depending on the specific tasks. They include:
> - Substitute from the reward a baseline term, which is a moving average (see section 3, Eq.2)
> - Reward clipping (see section 3, under Eq.2)
> - Use different values of T for different tasks (see “discussion” in section 4)
> - Use improved training algorithms (e.g. PPO) for more challenging tasks, and slightly adjust reward generation schemes (see “discussion” in section 4, and Appendix A.1).
>
> We have also revised the submission to disclose more details on how we make these improvements. We will make all code and models trained in this paper available for reproducibility.
>
> [1] Neural optimizer search with reinforcement learning. ICML 2017.
> [2] DARTS: Differentiable Architecture Search. Arxiv 1806.09055.

---

### Official Review · AnonReviewer2 · 2018-11-06

**Rating:** 6
**Confidence:** 3

**Review:**

The authors proposed an AutoLoss controller that can learn to take actions of updating different parameters and using different loss functions.

Pros
1. Propose a unified framework for different loss objectives and parameters.
2. An interesting idea in meta learning for learning loss objectives/schedule.

Cons:
1. The formulation uses REINFORCE, which is often known with high variance. Are the results averaged across different runs? Can you show the variance? It is hard to understand the results without discussing it. The sample complexity should be also higher than traditional approaches.
2. It is hard to understand what the model has learned compared to hand-crafted schedule. Are there any analysis other than the results alone?
3. Why do you set S=1 in the experiments? What’s the importance of S?
4. I think it is quite surprising the AutoLoss can resolve mode collapse in GANs. I think more analysis is needed to support this claim.
5. The evaluation metric of multi-task MT is quite weird. Normally people report BLEU, whereas the authors use PPL.
6. According to https://github.com/pfnet-research/chainer-gan-lib, I think the bested reported DCGAN results is not 6.16 on CIFAR-10 and people still found other tricks such as spectral-norm is needed to prevent mode-collapse.

Minor:
1. The usage of footnote 2 is incorrect.
2. In references, some words should be capitalized properly such as gan->GAN.

---

> ### Author Response · Authors · 2018-11-14
> **Thanks for the comments but some of your criticisms are invalid**
>
> We have fixed the footnote and capitalization problems. Below are replies to other comments.
>
> >> Comment #1
> We agree vanilla REINFORCE can exhibit high variance. However, as we have elaborated in the text below Eq.2, to reduce the variance and stabilize the training, we have made the following adaptations referring to previous works [1,2]:
> - Substitute a moving average B (defined in text) from the reward
> - Clip the final reward to a given range
> We empirically found the two techniques significantly stabilize the controller training.
> Moreover, AutoLoss is not restricted to REINFORCE, but open to any off-the-shelf policy optimization method, e.g. for large-scale tasks such as NMT, we introduce PPO to replace REINFORCE, and adjust the reward generation scheme accordingly (see the paragraph “Discussion”). We’ve also revised Appendix A.1 to cover details of how PPO is incorporated. Empirically, with random parameter initialization most experiments manage to converge and give fairly good controllers.
>
> Almost all main results are averaged over multiple runs as explicitly indicated in the main text and the table or figure captions (e.g. see captions of Table.1 and Fig.2). See Fig.2 and Fig.3(R) where vertical bars indicate variances. We have also updated Table.1 to show the variance.
>
> We will release all code and trained models for reproducibility.
>
> >> Comment #2
> We have provided substantial analysis and visualizations on what AutoLoss has learned in our *initial submission*. Below, we summarize them for your reference:
>
> - d-ary regression and MLP classification
> *See sec 5.1, the 3rd paragraph in P6 for analysis, and Table.1 for comparisons to handcrafted schedules*: we observe AutoLoss optimizes L1 whenever needed during the optimization. By contrast, linear combination objectives optimize both at each step while handcrafted schedules (e.g. S1-S3) optimize L1 strictly following the given schedule, ignoring the optimization status. We believe AutoLoss manages to detect the potential risk of overfitting using designed features, and combat it by optimizing L1 only when necessary.
> - GANs
> Per our observation, AutoLoss gives more flexible schedules than manually designed ones. It can determine when to optimize G or D by being aware of the current optimization status (e.g. how G and D are balanced) using its parametric controller.
> - NMT
> *See sec 5.1, the 3rd paragraph in P7 and Fig.3(M)*: we have explicitly visualized in Fig.3(M) the softmax output of a learned controller and explain in text: “...the controller meta-learns to up-weight the target NMT objective at later phase…resemble the “fine-tuning the target task” strategy...”.
>
> >> Comment #3
> We experimented with S>1 and found the improvement marginal. However, a large S requires more task model training steps to perform one PG (or PPO) update, meaning longer overall wallclock time for the controller to converge. We hence use S=1 as it performs satisfactorily. Note that some recent meta-learning literature uses policy gradient with batchsize 1, and report strong empirical results [3].
>
> >>  Comment #4
> We’d like to clarify that we have *not* claimed that “AutoLoss can resolve mode collapse in GANs”. AutoLoss improves the performance of GANs by enabling an adaptive optimization schedule than a pre-fixed one. Our point is better and faster convergence of the model training. In the GAN experiments we *qualitatively* observed the generated images are of satisfying quality and exhibit no mode collapse. But we never claimed we aim to or can resolve mode collapse.
>
> >> Comment #5
> We respectfully disagree with this comment. The NMT experiments aim to verify that AutoLoss can guide the multi-task optimization toward faster and better convergence on the target task, i.e. our interest is to see how the optimization goes instead of how the MT performs. Held-out PPL is the direct indicator of the quality of convergence, while BLEU evaluates the MT performance. Hence we believe PPL suffices as a metric to evaluate the performance of AutoLoss.
>
> >> Comment #6
> We acknowledge that there may exist DCGAN implementations that achieve higher IS on CIFAR-10, but note the following facts:
> - The link verifies in a table that the best official IS (reported in literature) is 6.16 (the number we report).
> - The self-implemented DCGAN 1:1 baseline used in our paper (see Fig.4(c)) achieves an IS=6.7, higher than 6.16.
> - Still, AutoLoss-guided DCGAN achieves IS=7, higher than 6.16 reported in literature, our own implementation, and the result from your link.
>
> Thanks again for mentioning spectral norm. However, these techniques are *completely orthogonal* from the scope of this paper, where we focus on whether AutoLoss can improve the convergence instead of resolving mode collapse.
> [1] Device Placement Optimization with Reinforcement Learning. ICML’17
> [2] Neural Optimizer Search with Reinforcement Learning. ICML’17
> [3] Efficient Neural Architecture Search via Parameter Sharing. ICML’18

---

> > ### Comment · AnonReviewer2 · 2018-11-26
> > **Thanks for clarifying the points, but stronger evaluation will be better.**
> >
> > Thanks the authors for addressing my comments. I’ve adjusted my score accordingly. I still think there are some weakness in terms of evaluation.
> > 1. IS is not the only qualitative metric in GAN and DCGAN is not the state-of-the-art baseline. I would be curious to see the how does AutoLoss perform using some more recent GAN architectures. In addition to IS, FID score is also a recent complimentary metric to show the effectiveness.
> > 2. I understand the response from comment 5, but reporting the metric that the community care about is also import. Sometimes, PPL is not directly correlated with BLEU or other indirect measure. Without reporting proper metrics, it is hard to know how the approach performed compared to Niehues & Cho 2017.

---

> > > ### Author Response · Authors · 2018-11-27
> > > **Thanks for the suggestion!**
> > >
> > > Thanks for the great suggestions again! We're working on generating new results using suggested metrics on the GAN and NMT experiments and will add the new results in the next version.

---

### Official Review · AnonReviewer4 · 2018-11-06
**Interesting idea. Clear paper.**

**Rating:** 7
**Confidence:** 4

**Review:**

Summary: This paper proposes a meta-learning solution for problems involving optimizing multiple loss values. They use a simple (small mlp), discrete, stochastic controller to control applications of updates among a finite number of different update procedures. This controller is a function of heuristic features derived from the optimization problem, and is optimized using policy gradient either exactly in toy settings or in a online / truncated manor on larger problems. They present results on 4 settings: quadratic regression, MLP classification, GAN, and multi-task MNT. They show promising performance on a number of tasks as well as show the controllers ability to generalize to novel tasks.

This is an interesting method and tackles a impactful problem. The setup and formulation (using PG to meta-optimize a hyper parameter controller) is not extremely novel (there have been similar work learning hyper parameter controllers), but the structure, the problem domain, and applications are. The experimental results are through, and provide compelling proof that this method works as well as exploration as to why the method works (analyzing output softmax). Additionally the "transfer to different models" experiment is compelling.

Comments vaguely in order of importance:
1. I am a little surprised that this training strategy works. In the online setting for larger scale problems, your gradients are highly correlated and highly biased. As far as I can tell, you are performing something akin to truncated back back prop through time with policy gradients. The biased introduced via this truncation has been studied in great depth in [3] and shown to be harmful. As of now, the greedy nature of the algorithm is hidden across a number of sections (not introduced when presenting the main algorithm). Some comment as to this bias -- or even suggesting that it might exist would be useful. As of now, it is implied that the gradient estimator is unbiased.

2. Second, even ignoring this bias, the resulting gradients are heavily correlated. Algorithm 1 shows no sign of performing batched updates on \phi or anything to remove these corrections. Despite these concerns, your results seem solid. Nevertheless, further understanding as to this would be useful.

3. The structure of the meta-training loop was unclear to me. Algorithm 1 states S=1 for all tasks while the body -- the overhead section -- you suggest multiple trainings are required ( S>1?).

4. If the appendix is correct and learning is done entirely online, I believe the initialization of the meta-parameters would matter greatly -- if the default task performed poorly with a uniform distribution for sampling losses, performance would be horrible. This seems like a limitation of the method if this is the case.

5. Clarity: The first half of this paper was easy to follow and clear. The experimental section had a couple of areas that left me confused. In particular:
5.1/Figure 1: I think there is an overloaded use of lambda? My understanding as written that lambda is both used in the grid search (table 1) to find the best loss l_1 and then used a second location, as a modification of l_2 and completely separate from the grid search?

6. Validation data / test sets: Throughout this work, it is unclear what / how validation is performed. It seems you performing controller optimization (optimizing phi), on the validation set loss, while also reporting scores on this validation set. This should most likely instead be a 3rd dataset. You have 3 datasets worth of data for the regression task (it is still unclear, however, what is being used for evaluation), but it doesn't look like this is addressed in the larger scale experiments at all. Given the low meta-parameter count of the I don't think this represents a huge risk, and baselines also suffer from this issue (hyper parameter search on validation set) so I expect results to be similar.

7. Page 4: "When ever applicable, the final reward $$ is clipped to a given range to avoid exploding or vanishing gradients". It is unclear to me how this will avoid these. In particular, the "exploding" will come from the \nabla log p term, not from the reward (unless you have reason to believe the rewards will grow exponentially). Additionally, it is unclear how you will have vanishing rewards given the structure of the learned controller. This clipping will also introduce bias, this is not discussed, and will probably lower variance. This is a trade off made in a number of RL papers so it seems reasonable, but not for this reason.

8. "Beyond fixed schedules, automatically adjusting the training of G and D remains untacked" -- this is not 100% true. While not a published paper, some early gan work [2] does contains a dynamic schedule but you are correct that this family of methods are not commonplace in modern gan research.

9. Related work: While not exactly the same setting, I think [1] is worth looking at. This is quite similar causing me pause at this comment: "first framework that tries to learn the optimization schedule in a data-driven way". Like this work, they also lean a controller over hyper-parameters (in there case learning rate), with RL, using hand designed features.

10. There seem to be a fair number of heuristic choices throughout. Why is IS squared in the reward for GAN training for example? Why is the scaling term required on all rewards? Having some guiding idea or theory for these choices or rational would be appreciated.

11. Why is PPO introduced? In algorithm 1, it is unclear how PPO would fit into this? More details or an alternative algorithm in the appendix would be useful. Why wasn't PPO used on all larger scale models? Does the training / performance of the meta-optimizer (policy gradient  vs ppo) matter? I would expect it would. This detail is not discussed in this paper, and some details -- such as the learning rate for the meta-optimizer I was unable to find.

12. "It is worth noting that all GAN K:1 baselines perform worse than the rest and are skipped in Figure 2, echoing statements (Arjovsky, Gulrajani, Deng) that more updates of G than D might be preferable in GAN training." I disagree with this statement. The WGAN framework is built upon a loss that can be optimized, and should be optimized, until convergence (the discriminator loss is non-saturating) -- not the reverse (more G steps than D steps) as suggested here. Arjovsky does discuss issues with training D to convergence, but I don't believe there is any exploration into multiple G steps per D step as a solution.

13. Reproducibility seems like it would be hard. There are a few parameters (meta-learning rates, meta-optimizers) that I could not find for example and there is a lot of complexity.

14: Claims in paper seem a little bold / overstating. The inception gain is marginal to previous methods, and trains slower than other baselines. This is also true of MNT section -- there, the best baseline model is not even given equal training time! There are highly positive points here, such as requiring less hyperparameter search / model evaluations to find performant models.

15. Figure 4a. Consider reformatting data (maybe histogram of differences? Or scatter plot). Current representation is difficult to read / parse.

Typos:
page 2, "objective term. on GANs, the AutoLoss: Capital o is needed.
Page 3: Parameter Learning heading the period is not bolded.

[1] Learning step size controllers for robust neural network training. Christian Daniel et. al.
[2]http://torch.ch/blog/2015/11/13/gan.html
[3] Understanding Short-Horizon Bias in Stochastic Meta-Optimization, Wu et.al.

Given the positives, and in-spite of the negatives, I would recommend to accept this paper as it discusses an interesting and novel approach when controlling multiple loss values.

---

> ### Author Response · Authors · 2018-11-14
> **Response to AnonReviewer4**
>
> Thanks for the detailed and encouraging feedback! We reply all comments below (relevant ones are put together):
>
> >> Comments #1, #11
> We mainly account the success of this simple training strategy to the simplicity of the model, the relatively low dimensionality of our input features, and the simplified action space (though all  three suffice to obtain a good controller in the current settings). They make the training of the controller much easier compared to other RL tasks with higher dimensional features or larger output space.
>
> We have added the detailed PPO-based training algorithm in Appendix A.1. While AutoLoss is amenable to different policy optimization algorithms, we empirically find PPO performs better on NMT, but REINFORCE performs better on GANs. As to the online setting, thanks for pointing us to the “short-horizon bias” paper. We have indicated in the revision the existence of this bias -- this bias was observed on the GAN task -- overtraining G can increase IS in a short term, but may lead to divergence in a long term as G becomes too strong. On the other hand, we didn’t observe it harms on NMT task noticeably. We hypothesize the tradeoff is insignificant on NMT, as in our multi-task setting, slightly over-optimizing one task objective usually does not have irreversible negative impact on the MT model (as long as the other objectives are optimized appropriately later on).
>
> >> Comments #2, #3
> We’d like to clarify that S=1 is consistent in the overhead section and Algorithm.1. S controls how many sequences to generate to perform a (batched) policy update (i.e. S is the batch size), and we set S=1 for all tasks. Only T differs across tasks, but we always update \phi whenever a reward is generated.
>
> Back to comment #2: for regression and classification, we have experimented with larger S and found the improvement marginal. As each reward is generated via an independent experiment, the correlations among gradients are unobvious. For large-scale tasks, we use memory replay to alleviate correlations in online settings (please see Algorithm 2 in Appendix A.1 in our revised version).
> Performing batched update with a larger S might help reduce correlations; However, a large S, as a major drawback, requires performing ST (S>>1) steps of task model training, in order to perform one step of controller update. This yields better per-step convergence, but longer overall training (wallclock) time for the controller to converge. There might exist sweet spots for S where one can achieve both good per-step convergence and short training time, but we skip the search of S and simply use S=1 as it performs well.
> It is worth noting that some recent literature uses a stochastic estimation of the policy gradient with batch size 1 as well, and report strong empirical results [1].
>
> [1] Efficient Neural Architecture Search via Parameter Sharing. ICML 2018
>
> >> Comment #4
> We observe the controller performance on all 4 tasks are insensitive to initialization. A good initialization (e.g. in NMT, equally assigning probabilities to each loss at the start of the training) indeed leads to faster learning, but most experiments with random initializations manage to converge to a good optima, thanks to \epsilon-greedy sampling used in training.
>
> >> Comment #5
> They are the same -- there is a typo leading to confusion in the sentence “...in Figure 1 where we set different \lambda in l_2 = \lambda |\Theta|_2...”; which should be “...in Figure 1 where we set different \lambda in l_2 = \lambda |\Theta|_1...”. We have fixed it in the latest version.
>
> >> Comment #6
> Please see the last paragraph in page 5. For regression, classification and NMT, we split data into 5 partitions D_{train}^C, D_{val}^C, D_{train}^T, D_{val}^T, D_{test}. AutoLoss uses D_{train}^C and D_{val}^C to train the controller. Once trained, the controller guides the training of a new task model on another two partitions D_{train}^T, D_{val}^T. Trained task models are evaluated on D_{test}. Baseline methods use the union of D_{train}^C, D_{val}^C, D_{train}^T, D_{val}^T for training/validation. For GANs that do not need a validation or test set, we follow the same setting in [1] for all methods.
>
> [1] Unsupervised Representation Learning with Deep Convolutional Generative Adversarial Networks. ICLR 2016.
>
> >> Comment #7
> Thanks for pointing out -- we apologize for misusing “exploding or vanishing gradients” and have revised the paper to be accurate. We simply intended to clip the reward to reduce variances, and fount it effectively improved training.

---

> > ### Comment · AnonReviewer4 · 2018-11-19
> > **Interesting**
> >
> > Thank you for your detailed comments.
> >
> > The addition of the appendix sections will greatly aid in reproducibility!
> >
> > @ Horizon bias: Interesting that you observe in GAN but not in MNT.
> >
> > One other small typo:
> >
> > A.8. Double reference to Algorithm 1 in GAN section. You probably mean one to be Algorithm 2.

---

> > > ### Author Response · Authors · 2018-11-27
> > > **Thanks for the comments!**
> > >
> > > Thanks for the comments again! We have fixed this typo in the latest version.

---

> ### Author Response · Authors · 2018-11-14
> **Response to AnonReviewer4  -- continued**
>
> >> Comment #8, #9
> Thanks for pointing us to these two works. In [1], the authors investigate several features and develop a controller that can adaptively adjust the learning rate of the ML problem at hand, similarly in a data-driven way. In [2], the authors propose to manually balance the training of G and D by monitoring how good G and D are, assessed by three quantities and realized by simple thresholding. By contrast, AutoLoss offers a more generic way to parametrize and learn the update schedule. Hence, AutoLoss fits into more problems (as we’ve shown in the paper).
> We have appropriately revised the two claims and cited them in the latest version.
>
> >> Comment #10
> Empirically, IS^2 or IS do not make much difference on the performance. The scaling term is a flexible parameter that controls the scale of the reward which we do not tune very much though.
>
> >> Comment #12
> Yes, in WGAN, it is preferable to train the critic till optimality. We have revised the statement for accuracy -- we observe in our experiments, for DCGANs with the vanilla GAN objective (JSD), more generator training than discriminator training generally performs better (but this may not be an effective hint for other GAN objectives as they behave very differently).
>
> >> Comment #13
> We have added Appendix A.8 to disclose all hyperparameters. All code and model weights used in this paper will be made available.
>
> >> Comment #14
> We’ve revised our statements to be more accurate: for all GANs and NMT experiments, we observe AutoLoss reaches better final convergence; For GAN 1:1, GAN 1:9, AutoLoss trains faster; for NMT experiments, AutoLoss not only trains faster but also converges better.
>
> We’d like to clarify that for all our GANs and NMT experiments, the stopping criteria of an experiment is either divergence or when we don’t observe improvement of convergence for 20 continuous epochs. This is why in Fig.2, Fig.3(L) and Fig.4(c), it looks like that different methods are given different training time.
>
> >> Comment #15
> We have update Figure.4(b) to a scatter plot, and fixed mentioned typos in the current version.

---

### Author Response · Authors · 2018-11-14
**Revision uploaded**

We thank all reviewers for giving valuable feedback to this paper. We have uploaded a revised manuscript in which we have incorporated the suggestions from the comments.

We want to highlight the following revisions:
- We have added to Appendix A.1 the detailed algorithm how PPO is incorporated into AutoLoss.
- Add Appendix A.8 to disclose detailed hyperparameters to produce the presented results.
- Add Appendix A.9 to discuss the potential limitations of AutoLoss, as suggested by AnonReviewer3.
- We have updated Figure.4(b) to a scatter plot for clarity, suggested by AnonReviewer4.
- We have added several references suggested by AnonReviewer4 and revised several claims to be more accurate.

---

### Public Comment · ~Fei_Tian1 · 2018-12-17
**Thank you for referring to L2T**

Dear the authors,

Thank you for referring to our ICLR'18 work "Learning to Teach" in your work. We have an extension of L2T in NeurIPS this year: "Learning to Teach with Dynamic Loss Functions" (https://papers.nips.cc/paper/7882-learning-to-teach-with-dynamic-loss-functions.pdf), which studies the automatic discovery of better objectives/loss functions adaptively in the optimization process, and therefore is quite related with your work. It'll be more comprehensive to position this one in your paper. Thanks.

Best,
Fei Tian

---

> ### Author Response · Authors · 2018-12-20
> **Thank you for pointing us to your work and we will cite your paper in a future version**
>
> Thank you for pointing us to your work [1], which studies the similar topic concurrently with us. Both works focus on designing methods to introducing dynamicas into objectives/loss functions. Specifically, [1] tries to directly cast the objective function as a a learnable neural network (learned by measuring the similarity between model prediction and ground-truth). By contrast, we focus on learning the update schdules (parameterized as NNs) in problems where multiple objectives or/and sets of parameters are involved. Our formulation allows for tackling alternate optimization problems such as (1) GANs, where multiple objectives have clear difference with each other and are combined in a minimax form; (2) multi-task learning, that each objective of interest is well-defined and prefixed but an update order is missing; (3) or even EM-based maximum likelihood estimation where some inference procedures involved (e.g. MCMC) aren't in the form of a gradient-based optimization -- In all these cases, the objective itself might be difficult to be represented or approximated by neural networks. We will cite your paper in a future version and include the above discussion.
>
> [1] Wu, L., Tian, F., Xia, Y., Fan, Y., Qin, T., Jian-Huang, L., & Liu, T. Y. (2018). Learning to Teach with Dynamic Loss Functions. In Advances in Neural Information Processing Systems (pp. 6465-6476).

---

### Meta-Review · Area_Chair1 · 2018-12-14
**Accept**

**Confidence:** 4
**Recommendation:** Accept (Poster)

**Metareview:**

The paper suggests using meta-learning to tune the optimization schedule of alternative optimization problems. All of the reviewers agree that the paper is worthy of publication at ICLR. The authors have engaged with the reviewers and improved the paper since the submission. I asked the authors to address the rest of the comments in the camera ready version.